# Green Synthesis of Laser-Induced Graphene with Copper Oxide Nanoparticles for Deicing Based on Photo-Electrothermal Effect

**DOI:** 10.3390/nano12060960

**Published:** 2022-03-14

**Authors:** Jun-Uk Lee, Jeong-hoon Lee, Chan-Woo Lee, Su-Chan Cho, Sung-Moo Hong, Yong-won Ma, Sung-Yeob Jeong, Bo-Sung Shin

**Affiliations:** 1Department of Cogno-Mechatronics Engineering, Pusan National University, Pusan 46241, Korea; lju3534@naver.com (J.-U.L.); kgslsk6219@naver.com (J.-h.L.); cwleeho2@naver.com (C.-W.L.); cho_brian@naver.com (S.-C.C.); 2Interdisciplinary Department for Advanced Innovative Manufacturing Engineering, Pusan National University, Pusan 46241, Korea; hsm14789@naver.com (S.-M.H.); decentsoul@naver.com (Y.-w.M.); 3Department of Mechanical Engineering, The University of Tokyo, Tokyo 113-8656, Japan; ysjsykj8025@naver.com; 4Department of Optics and Mechatronics Engineering, Pusan National University, Pusan 46241, Korea

**Keywords:** flexible, laser-induced graphene, polyimide, deicing, photothermal effect, heater

## Abstract

Homogenously dispersed Cu oxide nanoparticles on laser-induced graphene (LIG) were fabricated using a simple two-step laser irradiation. This work emphasized the synergetic photo-electrothermal effect in Cu oxide particles embedded in LIG. Our flexible hybrid composites exhibited high mechanical durability and excellent thermal properties. Moreover, the Cu oxide nanoparticles in the carbon matrix of LIG enhanced the light trapping and multiple electron internal scattering for the electrothermal effect. The best conditions for deicing devices were also studied by controlling the amount of Cu solution. The deicing performance of the sample was demonstrated, and the results indicate that the developed method could be a promising strategy for maintaining lightness, efficiency, excellent thermal performance, and eco-friendly 3D processing capabilities.

## 1. Introduction

There have been growing concerns about the accumulation of ice owing to the increasing use of electronic instruments, aircraft, power grids, and telecommunications, which leads to many problems in the industry [1,2,3,4]. Furthermore, ice accumulation can cause immediate safety problems while driving an electronic car or boarding an airplane [5,6]. Therefore, developing rapid deicing devices is essential to help maintain the performance of electronic products, especially in the aerospace industry. In recent years, tremendous efforts have been devoted to finding materials for deicing, such as electroconductive textiles, carbon nanotubes (CNTs), sprayable metal layers, and carbon fibers (CFs) [7,8,9,10]. Among these, highly efficient, lightweight, and flexible materials are vital for applications based on electronic devices. Recently, graphene has received great attention because of its large surface area, good thermal conductivity, excellent mechanical strength, and superior electronic mobility [11,12,13].

Graphene, a single-atom thick two-dimensional honeycomb lattice of $sp^{2}$ carbon allotropes is a versatile material that offers high photothermal conversion efficiency, tunable wettability, high mechanical strength, and ease of functionalization [14,15,16]. Recently, significant effort has been devoted to graphene-based deicing devices. Graphene-based devices could efficiently heat up by combining the photo/electro-to-heat effect of graphene for highly efficient deicing devices. While graphene-based devices have unique properties, their applications are limited by complicated, expensive, and inefficient manufacturing routes and are not suitable for industrial applications [17,18,19,20]. Laser-induced graphene (LIG) could be a good strategy to fabricate graphene-based composites [21,22]. LIG is produced by directly irradiating carbonaceous precursors, and it naturally exhibits a three-dimensional porous structure. LIG and its derivatives have broad applications, such as sensors, batteries, and catalysts [23,24,25,26], because of their simple, time-efficient, low-cost, and eco-friendly preparation using an expanding range of raw materials.

In this paper, we report a facile and efficient strategy to engineer polyimide (PI) and metal solutions into metal oxide nanoparticles with LIG for a deicing device using a two-step irradiation method. This method uses a 355 nm UV pulsed laser to directly convert PI into hydrophilic LIG and fabricate Cu oxide nanoparticles. By regulating the amount of the metal solution, we can readily control the Cu oxide nanoparticle configuration. The proposed method allows facile fabrication of durable deicing devices with a high photo-electrothermal effect.

## 2. Materials and Methods

### 2.1. Laser Delivery System

A commercial 355 nm pulsed laser system (AONano 355-5-30-V from Advanced Optowave, Ronkonkoma, NA, USA) was used as the laser source for fabricating the LIG. An Nd:YVO_4_ laser with a maximum average power of 5 W at λ = 355 nm, a pulse length of τ = 15 ns, and repetition rate of ƒ = 30 kHz was used to irradiate the samples in a standard atmospheric environment (25 °C and normal pressure), as described in Appendix A. Laser direct writing (LDW) was adopted for the preparation of LIG patterns during laser induction under ambient conditions at room temperature. Laser beam delivery was performed by moving the mirrors of a Galvano scanner (HurrySCAN III 14 from SCANLAB, Pucheim, Germany) and the F-θ lens of focal length f = 105.9 mm (S4LFT4100/075 Telecen-tric Scan Lens from Sill Optics, Wendelstein, Germany).

### 2.2. Preparation of LIG and Cu Oxide Nanoparticles on LIG

The fabrication process is illustrated in Figure 1. The laser path of the 355 nm UV pulsed laser is shown in Figure 1a. The laser power was 1.1 W, the laser scan mode was unidirectional, and the laser scanning speed was 60 and 40 mm/s. Appendix A shows the laser beam condition. The weak laser power prevents the excessive oxidation of LIG, which causes corrosion and weakens the photo-electrothermal effect. The other experimental details are described in the Supporting Information. The laser beam was scanned using a crosshatching process to form grooved LIG patterns with a porous structure. The hatch distance was fixed at 0.05 mm. Both the horizontal and vertical scan modes provided a high porosity to the surface. Appendix A shows the LIG patterns fabricated using only the horizontal scan mode (0°). The porosity of this pattern was fabricated at the laser spot area. As shown in Figure 1b, the LIG patterns were directly fabricated with the laser scanning speed of 60 mm/s on a commercial polyimide (PI) film with a thickness of 125 μm (Kapton^®^ from DupontTM Wilmington, DE, USA). Figure 1e shows the FEM image of Porous LIG. Then, 1% CuCl_2_ solution (Sigma-Aldrich, Saint Louis, MO, USA) was dropped on the LIG patterns for the formation of Cu oxide nanoparticles. The LIG sample containing the Cu solution was subsequently re-irradiated with the 355 UV pulsed laser under the laser speed of 40 mm/s, as shown in Figure 1c. Figure 1d shows the actual LIG with Cu oxide nanoparticles. For convenience, we refer to this material as Cu/LIG. Figure 1f shows the FE-SEM image of the Cu/LIG. This has porous structures nanoparticles on the surface.

### 2.3. Characterization

The morphology of the LIG was studied using field emission scanning electron microscopy (FE-SEM, TESCAN MIRA 3 LMH In-Beam Detector, Brno, Czech Republic). Raman spectra were measured using a Raman spectrometer (NRS-5100 JASCO International Co., Ltd., Tokyo, Japan) with a 532-nm excitation line. X-ray photoelectron spectroscopy (XPS) was also performed to further analyze the surface chemical compositions of LIG and Cu/LIG. The instantaneous electrothermal performance was recorded using an LCR meter 4100 (Wanye Kerr Electronics, Woburn, MA, USA) and a Keithley 2450 source meter (Keithley Tektronix, Beaverton, OR, USA). Moreover, the zeta potentials were analyzed using a Litesizer 500 (Anton Paar, Graz, Austria), and the surface area and pore size were recorded using an Autosorb-iQ instrument (Quantachrome, Boynton Beach, FL, USA).

## 3. Results

### 3.1. Morphological Characteristics of Cu/LIG

For the fabrication of graphene-based metallic hybrid materials, many methods are attempted, such as reduction, liquid-phase exfoliation, and electrodeposition [27,28,29]. These methods can produce metal hybrid graphene with excellent performance, but they require harmful chemicals or an environment that applies high pressure and heat. We fabricate Cu oxide nanoparticles on the LIG surface very quickly and easily using LDW without harmful chemicals under ambient conditions.

Figure 2 shows the FE-SEM images of the Cu/LIG patterns according to Cu solution varying 2.5, 5, and 7.5 μL. Figure 2a,d,g shows the patterns with 2.5 μL Cu solution. Cu oxide nanoparticles on LIG could be shown, but their distribution density looks lower. It is analyzed that the amount of solution is not enough to spread Cu oxide nanoparticles on the LIG surface. Figure 2b,e,h shows the patterns fabricated at an amount of 5 μL. Their highly distributed Cu oxide nanoparticles show that the 5 μL Cu solution is the best condition for Cu/LIG samples. Their Cu oxide nanoparticles could be used as a synergetic effect for photo-electrothermal effect with LIG. Figure 2c,f,i shows Cu/LIG fabricated at the amount of 7.5 μL. Owing to the relatively higher amount of Cu solution, the energy of fluency of second irradiation could be used as vaporizing of water solution, which weakens fabrication of Cu oxide nanoparticles.

### 3.2. Chemical Characteristics of Cu/LIG

The Raman spectra of LIG Cu/LIG show three characteristic peaks located at 1346, 1584, and 2695 cm^−1^, which belong to the D, G, and 2D peaks of the graphene-derived material, as shown in Figure 3a,b [30,31]. The D peak was caused by the defects or disordered $sp^{2}$ carbon, the G peak was induced by the $sp^{2}$ carbon, and the 2D peak originated from the second-order zone-boundary phonons. Both spectra have sharp, symmetrical 2D peaks, providing that graphene is indeed produced during the LIG process [31].

If the $I_{g}$/$I_{2D}$ ratio is greater than 1, then the graphene is multilayered. Because the experimental $I_{g}$/$I_{2D}$ ratio was 1.13, it was considered as a multilayered structure. In addition, the D peak was much stronger than that of conventional graphene [31]. This is because the LIG contains both graphene oxide and reduced graphene oxides.

Compared with the bare LIG, the ratio of the D peak gradually increased in Cu/LIG, as shown in Figure 3b. The $I_{D}$/$I_{G}$ ratio of LIG was approximately 0.97, and that of Cu/LIG was approximately 1.01. Thus, dangling bonds and substitution of Cu elements through the catalytic reaction of aqueous copper chloride can lead to defective LIG. In addition, the peaks near 300 cm^−1^ and 610 cm^−1^ are copper oxide peaks [32]. Meanwhile, Figure 3c shows the XPS survey spectra of LIG and Cu/LIG, which show that carbon and oxygen are the main peaks. As shown in Figure 3d,e, the C1s region for LIG and Cu/LIG was deconvoluted into three peaks at 284.6, 286, and 290 eV, corresponding to C–C bonds with $sp^{2}$ hybridization, C–O bonds, and C=O bonds, respectively. Their full width at half maximum (FWHM) indicates the ratio of the C1s peak to the O1s peak. The relatively lower FWHM of Cu/LIG compared with LIG indicates that the second irradiation with Cu solution reduces the oxygen functional groups [33,34]. The XPS spectra for Cu/LIG were also examined in the Cu2p region, as shown in Figure 3f. The binding energies of Cu2p3/2 and Cu2p1/2 peaks were located at 933 eV and 952.3 eV, respectively [35]. The broader Cu2p peaks show that Cu/LIG has Cu nanoparticles of oxide foam rather than metallic foam. As shown in Appendix A, EDS spectra of Cu/LIG show Cu oxide particles have abundant oxygen groups. Appendix A shows the EDS spectra table of Cu/LIG. The strong Cu2+ satellite peaks located at 942.7 eV and 962.8 eV show that the Cu oxide nanoparticles appear to be CuO [36]. Appendix A shows the TGA curves of Cu/LIG with enhanced thermal stability performance.

### 3.3. Photothermal Properties of Cu/LIG

The excellent photothermal properties of Cu/LIG samples were observed, as shown in Figure 4. We graphed the data as the average value of the results of three or more experiments. Furthermore, we studied the temperature change of the sample surface according to the solar power (SUN), as shown in Figure 4a,b. Figure 4a shows the temperature as a function of solar power for LIG patterns fabricated according to the laser scanning speed. The solar generator supplied power from 0.5 to 1.2 sun in 0.1 increments. Each prepared sample was exposed for 3 min, and the temperature was measured using a thermal imaging camera. All samples showed a linear temperature increase from 0.5 to 1.1 sun. However, the difference temperature presented in Figure 4a is very small. We measured the surface temperature of the Cu/LIG surface according to the Cu solution from 0 to 7.5 µL in 2.5 increments, as shown in Figure 4b. LIG (1) and (2) showed the lowest temperature rise in all the samples, which confirmed that copper oxide nanoparticles improved the photothermal conversion properties of the LIG. Among the three Cu/LIG samples, Cu/LIG (5 µL) showed the highest temperature increase. Thus, the higher the Cu oxide particle content on the LIG surface, the larger the photothermal conversion. These results are consistent with the FE-SEM images showing that the amount of Cu oxide nanoparticles is largest at Cu/LIG (5 µL).

We compared the heating and cooling time of LIG (2) and Cu/LIG (5 µL), as shown in Figure 4c. This graph shows the time required to reach the maximum temperature at 1 sun and the cooling time when the solar generator was turned off. As the temperature gradient changes of LIG (2) and Cu/LIG (5 µL) were similar, the rising speed was also similar. We confirmed that the temperature difference was approximately 8 °C until the solar generator was turned off when both samples showed a rapid temperature drop. In the case of the Cu/LIG (5 µL) sample, the cooling time was slightly delayed compared with that of the LIG (2) sample. The cooling time of LIG was approximately 32 °C after 150 s, whereas that of Cu/LIG (5 µL) was approximately 33 °C after 300 s. This shows that LIG with Cu oxide nanoparticles produces heat more efficiently than bare LIG through photothermal conversion [37].

Figure 4d shows the surface temperature according to the area change from 1 × 1 cm^2^ with 5 µL, 2 × 2 cm^2^ with 20 µL, and 3 × 3 cm^2^ with 45 µL, which are the best conditions for photothermal conversion. The samples were exposed to 1 sun for 5 min and showed temperature changes of 78.6 °C, 81.7 °C, and 84 °C, respectively. We confirmed that the surface area affected the photothermal conversion. However, the temperature difference between 1 × 1 and 2 × 2 cm^2^ is 3.1 °C, and the difference between 2 × 2 and 3 × 3 cm^2^ is 2.3 °C. As the area increases, the difference in temperature conversion ability is expected to be insignificant, and it is expected that laser-induced heat will influence the fabricated smaller areas.

Figure 5a is a state in which 0.5 mm of water is frozen in the form of a column with a diameter of 0.8 cm to check the photothermal deicing performance of our Cu/LIG (5 µL) pad with a 3 × 3 cm^2^ area. The deicing tests were conducted by exposing the sample to a solar lamp at 1 sun from the ground. As soon as the pad is exposed to the light source, the interface between the LIG and ice is immediately melted, as shown in Figure 5b. The heat through photothermal conversion started to melt the ice gradually from the interface, and a water layer is eventually formed at the weak interface. Then, ice and LIG were separated by the formed water layer. As shown in Figure 5c, the water layer flows to the column by gravity and forms a droplet further away from the interface by surface tension [38]. At 93 s, the entire interface is melted, and the ice column falls to the bottom owing to gravity, as shown in Figure 5d. Figure 5e shows a schematic illustration of Cu/LIG of the deicing based on the photothermal effect. The excellent photothermal performance of our deicing device could be attributed to the three advantages of Cu/LIG: π-bonds, Cu oxide nanoparticles, and 3D porous structure. The general heat generation mechanism of carbonaceous materials is mainly through the thermal vibration of molecules. After absorbing incident solar light, carbon materials tend to be in an excited state with high-frequency phonon modes and then turn to the low-frequency modes via a phonon–phonon coupling process to release the energy, thus generating heat energy [39,40]. To enhance the photothermal efficiency, the molecular structure of LIG has numerous conjugated π-bonds that are different from those of conventional amorphous carbon. Configuration design plays a crucial role in light-to-heat efficiency. In addition to improving solar absorption, reducing the losses of incident light is another encouraging technology route to boost the photothermal property [41]. When incident light is irradiated onto the surface of the LIG, light reflection is usually accompanied by light absorption [42]. The reflected light will return to the air and thus cannot be effectively utilized if the surface of the deicing device is planar [43]. Taking these into account, reducing or reusing the reflected light is an effective approach to improve the deicing performance [44]. Hybridization of photothermal materials has also been demonstrated to be a powerful tool to obtain satisfactory light-to-heat performance. This approach takes advantage of the optical response of different solar absorbers to maximize the optical properties in both absorption range and intensity, based on multiple internal reflections at the interface owing to impedance mismatch, which then leads to photoabsorption. The interface of Cu oxide nanoparticles and LIG leads to enhanced optical absorption and reinforced hot electron–phonon interactions for efficient photothermal conversion [45]. Figure 5e shows the thermal images of 3 × 3 cm^2^ LIG and Cu/LIG pad at the maximum temperature of 1 sun. The maximum temperatures of LIG (2) and Cu/LIG (5 µL) were 80.9 °C and 88.3 °C, respectively. The Cu oxide particles showed an improved photothermal change of approximately 7.4 °C. Therefore, the excellent deicing performance of our Cu/LIG samples was explained by the hybridization of Cu oxide nanoparticles on porous LIG.

### 3.4. Electrothermal Performance Test

Figure 6 shows the electrical and electrothermal properties of the samples. The most important parameters in Joule heating are the resistance and current within the materials. Therefore, the I–V curve in Figure 6a is used to assess the electrical properties of the deicing device. The LIG and Cu/LIG (5 µL) show both linear I–V curves, which means that both samples have good ohmic behavior and can be controlled with applied voltage for electrothermal devices [46]. As shown in Figure 6b,c, we tested the linearity between the input voltage and temperature. We raised the temperature to 200 °C for safe operation considering the glass transition temperature of polyimide [47]. Cu/LIG heater has better temperature stability and linearity than LIG, according to varying voltage. Likewise, Cu/LIG shows a higher temperature change when applying the same voltage. These results indicate that the second irradiation with Cu solution increases the electrical conductivity of Cu/LIG patterns [48]. Meanwhile, power density is an index for the energy efficiency of the heater. It is the energy consumption per unit area in the case of a two-dimensional system. Thus, achieving a higher temperature at the same power density corresponds to a higher heater efficiency. Figure 6d shows the energy efficiency of the two kinds of heaters we manufactured. The Cu/LIG heater has much higher temperatures than bare LIG in all operating power densities. We found that the temperature difference widens as the power density increases. Our Cu/LIG heater showed very efficient heating performance with 0.27 W/ cm^2^ at 100 °C through a 1 cm^2^ scale. Compared with previous research, which showed a temperature of about 100 °C at a power density of 0.35 W/ cm^2^, our heater needed about 18.5% lower power density to reach the same temperature [49].

Furthermore, we found that the threshold voltages at 200 °C were 4 V and 6 V, respectively. Then, we measured the rising time to reach 200 °C, as well as the cooling time. As shown in Figure 6e,f, we confirmed that Cu/LIG decreases both the rising time and cooling time. The Cu/LIG heater reached the target temperature within 5 s, and its cooling time was just 13 s. Therefore, copper oxide can realize a rapid heater. We applied our Cu/LIG heater to the deicing.

Aircraft icing refers to the formation of ice on the surface of an aircraft due to cloud particles or supercooled water droplets below the freezing point temperature [50]. Supercooled water droplets are most often observed between 0 °C and –20 °C [51]. Therefore, we conducted a deicing test while maintaining an environment at –10 °C. Appendix A shows the process of the electrothermal performance test for deicing. First, 20 mm thick ice was formed on a 3 × 3 cm^2^ heater, which was attached above a cold metal to compensate for the real condition. Because of the heat sink and cold environment, we input a power density of 1.06 W/cm^2^. Figure 7a describes in detail how ice is finally vaporized. First, there is a state in which no voltage is applied to the heater. When voltage is applied, it undergoes rapid heating, which immediately creates water at the interface between the ice and heater. Water can reduce the adhesion of the ice and heater interface [52]. This means that ice can be removed easily by external forces such as wind and gravity. The third schematic shows that most of the heat is used to melt the ice again, but the edge side far from the ice undergoes evaporation. In this step, some of the moisture penetrates the 3D porous structure of the microscale. When whole ice is melted, as in the fourth image, all the heat is used to vaporize the moisture on the surface. Then, the temperature of the heater surface rises rapidly. Through this process, the heater was transformed into a completely dried state. This additional process must prevent icing because the remaining moisture on the surface subsequently acts as a frozen seed [53].

Figure 7b shows the change of the highest temperature point on the surface. As soon as the heater is turned on at −10 °C, the edge side is melted, and the temperature rises to about 50 °C. After that, the temperature rapidly increases continuously to 80 °C until 40 s. Then it faces an evaporation state. The temperature slowly rises from 40 s to 135 s.; the reason why it does not go over 102 °C is the boiling point of water. After the evaporation process, the temperature rises about 135 ° during 240 s.

## 4. Conclusions

In conclusion, we attempted to grow Cu oxide nanoparticles on a LIG surface using only a UV nanosecond pulsed laser. After optimizing the best Cu/LIG process through a photothermal experiment, the configuration of the Cu/LIG composites was studied by structural and morphological analyses. The hybrid Cu/LIG composites showed synergistic photo-electrothermal effects for deicing. We also analyzed the chemical characteristics using Raman spectroscopy. The photo and electrothermal effect of LIG was further improved through Cu oxides nanoparticles on the LIG surface based on light trapping and multiple electron internal scattering. By optimizing the photothermal effect with the number of Cu solutions, we found that a 5 µL Cu solution is suitable for a 1 × 1 cm^2^ area for Cu/LIG deicing devices. This work presents an eco-friendly, sustainable, and cost-effective fabrication pathway for potential deicing devices. We hope that this strategy could be used as an industrially viable route for the mass production of high-performance deicing devices.

## Figures and Tables

**Figure 1 nanomaterials-12-00960-f001:**
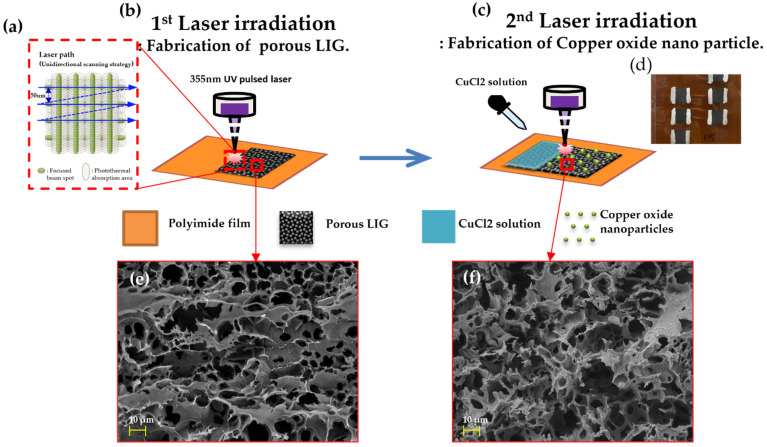
Fabrication of process of LIG deicing systems: (**a**) illustration of laser pulse spot; (**b**) first irradiation onto PI for porous LIG; (**c**) second laser irradiation with CuCl_2_ solutions for the fabrication of Cu oxide nanoparticles; (**d**) actual photograph of Cu/LIG with silver paste electrode; FE-SEM images of (**e**) porous LIG and (**f**) Cu/LIG.

**Figure 2 nanomaterials-12-00960-f002:**
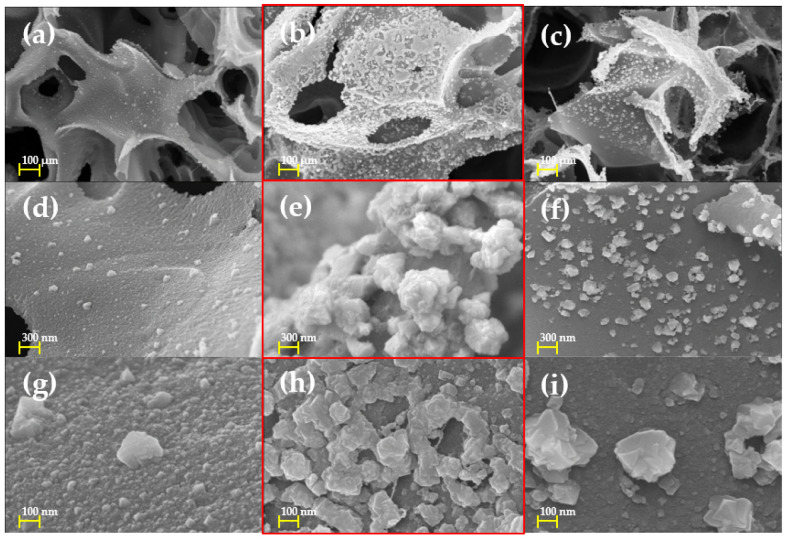
FE-SEM images of Cu/LIG patterns fabricated with varying amounts of Cu solution: (**a**,**d**,**g**) 2.5 µL; (**b**,**e**,**h**) 5 µL; (**c**,**f**,**i**) 7.5 µL.

**Figure 3 nanomaterials-12-00960-f003:**
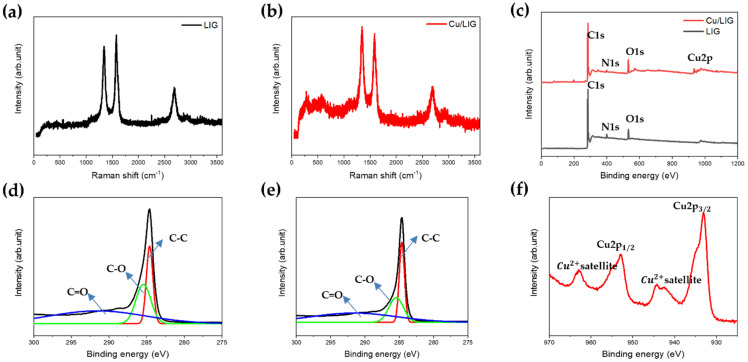
Raman spectra of (**a**) LIG and (**b**) Cu/LIG. (**c**) XPS characteristics of LIG and Cu/LIG, XPS C1s data of (**d**) LIG, (**e**) Cu/LIG, (**f**) XPS spectra for Cu 2p of Cu/LIG.

**Figure 4 nanomaterials-12-00960-f004:**
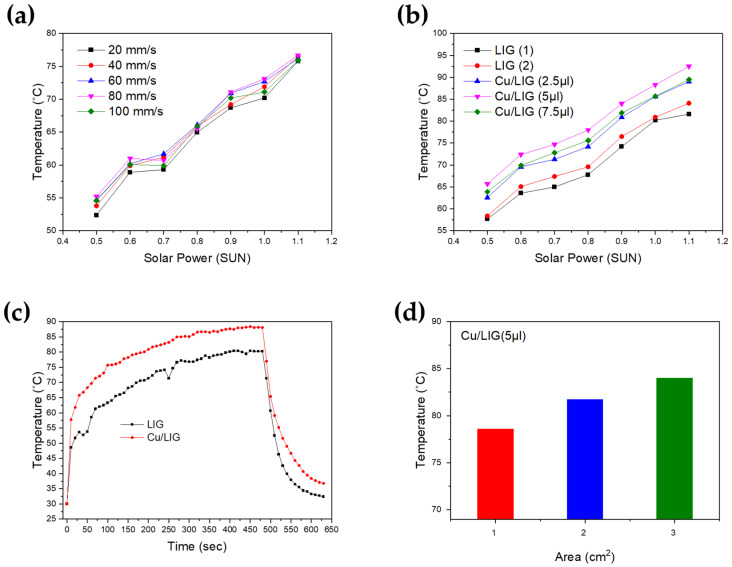
(**a**,**b**) Temperature as a function of solar power for LIG patterns, (**c**) photothermal rising and cooling time of LIG and Cu/LIG, (**d**) sample area-dependence of the temperature.

**Figure 5 nanomaterials-12-00960-f005:**
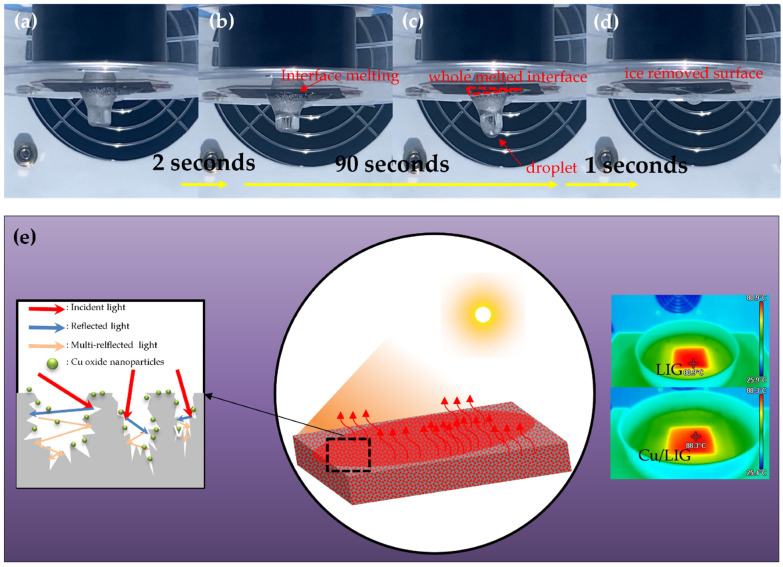
(**a**–**d**) Actual images of deicing based on photothermal effect with Cu/LIG fabricated with 5 µL of Cu solution. (**e**) Schematic illustration of the deicing mechanism based on photothermal effect and thermal image of the saturated temperature of LIG and Cu/LIG with 1 Sun.

**Figure 6 nanomaterials-12-00960-f006:**
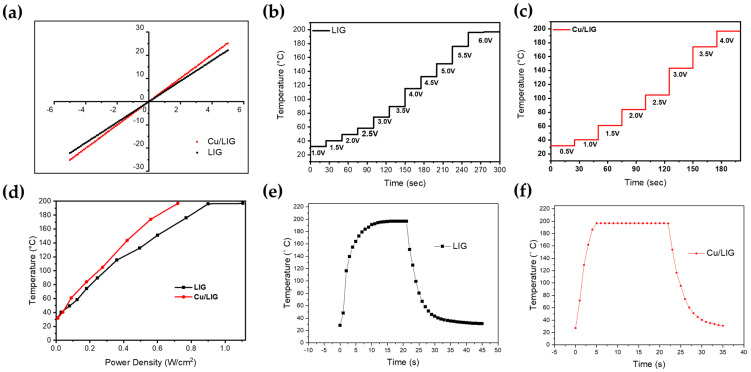
(**a**,**b**) I–V curve measurement of LIG and Cu/LIG. (**b**,**c**) Temperature evolution of LIG and Cu/LIG at stepwise voltage rise from 1 to 6 V. (**d**) Temperature as a function of the applied electrical power density for LIG and Cu/LIG. (**e**,**f**) Heating and cooling time for LIG and Cu/LIG.

**Figure 7 nanomaterials-12-00960-f007:**
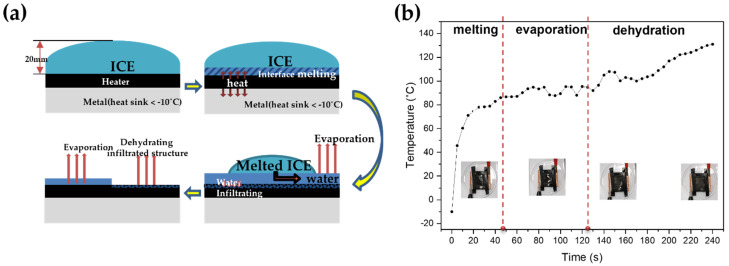
(**a**) Illustration of the deicing process based on electrothermal effect, (**b**) temperature graph of deicing with Cu/LIG heater.

## Data Availability

Not applicable.

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
