# Peer review of "Green Synthesis of Laser-Induced Graphene with Copper Oxide Nanoparticles for Deicing Based on Photo-Electrothermal Effect"

_nanomaterials, 2022, doi:10.3390/nano12060960_

Round 1

Reviewer 1 Report

Jun-Uk Lee et al. have prepared Cu oxide nanoparticles on a LIG surface via a UV nanosecond pulsed laser. The as-prepared sample exhibits synergistic photo-electro-thermal effects for de-icing. Moreover, the corresponding mechanism is also discussed. It is novel and a topic of interest to the researchers in the related areas. There’s still some minor revision that I suggest for it’s publication as seen below.

  1. The error bar should be given in Fig. 4b.
  2. The caption in Fig. 3 and 6 is too small to see, please enlarge it.
  3. The inset of Fig. 5e shows the Schematic illustration of cross profile. Therefore, the cross profile of the as-prepared surface should be given, just like LCM.
  4. More related-experimental papers, regarding on laser fabrication and its applications, may be introduced, for example: Applied Physics Letters, 2021, 118 (21), 211905
  5. The authors should polish their writing. There are some grammar mistakes throughout the paper.

Author Response

Dear reviewer,

  1. The error bar should be given in Fig. 4b.

There is no error bar because the saturation temperature according to the area is measured once. sorry. thanks for the good point

  1. The caption in Fig. 3 and 6 is too small to see, please enlarge it.

As you said, we increased the size of the units in the graph from 22 to 26.

  1. The inset of Fig. 5e shows the Schematic illustration of cross profile. Therefore, the cross profile of the as-prepared surface should be given, just like LCM.

It's a little hard to understand, so I took an actual thermal image and placed it on the right. I hope you can understand a little bit. Thanks for pointing it out.

  1. More related-experimental papers, regarding on laser fabrication and its applications, may be introduced, for example: Applied Physics Letters, 2021, 118 (21), 211905

Thanks for pointing out a good journal. I will read this journal and add it to my reference. thank you.

  1. The authors should polish their writing. There are some grammar mistakes throughout the paper.

The overall content was refined according to the advice of other reviewers. I would really appreciate it if a reviewer would check it again after seeing it. The quality of my thesis has improved a lot through the previous 5 review feedbacks. thank you.

Best regards,

JUNUK LEE

Reviewer 2 Report

Dear Authors,

the paper is well written and presented and there is something interesting in the proposal and in the results. However there are some methodological flaws and mistakes and the paper should be amended before publication.

  1. The main concern is that there is a small difference among the LIG and Cu/LIG based device performance and a sample statistical analysis I not presented.
  2. Differences between devices can be due to differences electrical resistance, which can be due to differences in geometry and in resistivity (materials properties). Neither the geometry nor the material properties were sufficiently characterized to justify the lengthy discussion presented in the paper.

Let’s see in details:

  1. Introduction: Please comment that LIG is not really Graphene but just a conventional name for mainly 2D coordinated nanostructured carbon, as it is clearly demonstrated by Raman in figure 3
  2. Materials and methods: For easier reading please add the energy per pulse and peak power to table s1
  3. Materials and methods: “The porosity of this pattern was fabricated at the laser spot area”. The pattern was porous at....
  4. Results: "For fabrication of graphene based metallic hybrid materials, many methods...." Please explain briefly why the present method is better.
  5. Results: Figure 2 is difficult to understand. According to the text different rows (a-c, d-f, g-h) were made with different CuCl2 amounts, but they are presented at different magnifications. Moreover the substrate geometry on the first column looks different from the second and third. It is difficult to make sense of it. Probably the marker were misplaced, higher MAG from left to right and not from top to bottom
  6. Results, below figure 3: “The number of graphene layers can be identified by the ratio of the intensities of the 2D and G peaks” This statement is not correct for highly defective graphene, there are many papers about it Thickness of carbon nanostructures should be determined using TEM or SEM. (L. G. Cançado, A. Jorio, E. H. M. Ferreira, F. Stavale, C. A. Achete, R. B. Capaz, M. V. O. Moutinho, A. Lombardo, T. S. Kulmala, A. C. Ferrari, Nano Letters 2011, 11, 3190.A. Eckmann, A. Felten, I. Verzhbitskiy, R. Davey, C. Casiraghi, Physical Review B 2013, 88, 035426. )
  7. Results: In the survey spectrum presented in figure 3 c, in the red curve it is possible to see also the Cl 2p peak at 200eV due to its incomplete removal. This peak should be also analysed.
  8. Results below figure 4: “from 0.5–1.2 sun”, probably 0.5 to 12
  9. Whole section: The differences presented in figure 4 are very small and are apparently meaningless unless a statistical analysis of at least 3 samples per type is presented.
  10. Results above figure 5: “This shows that Cu oxide not only produces heat more efficiently than bare LIG through photothermal con-version, but also provides heat resistivity to preserve heat [37].”: Please define what heat resistivity is, is it a decrease of thermal conductivity?
  11. Results page 8: “These increased conjugated bonds result in a red shift of the absorption light spectrum, which greatly benefits the solar energy utilization efficiency” This statement should be demonstrated experimentally, by presenting absorption and reflection curves (using integrating spheres), else dropped
  12. Results last 8 lines of 3.3 page 8 “the synergistic….” Overall the enhancement in performances seems too small to justify a different mechanism. Quite simply, the porous structure is obviously more absorbent (due to multiple reflections), and CuO particles are also black. The discussion is confusing and not supported by experimental evidence.
  13. Results 3.4 “However, the minimum operating volt-age of Cu/LIG heater in Figure 6c is about 0.5 V, which shows a 50% lower switching voltage than that of LIG heater.” This sentence is unclear, it should be either removed or better explained. If the IV curve is linear (figure 6a), there is no switching voltage and both devices can be operated at any voltage! The steps in figure 6b and 6c are arbitrary
  14. Figure s4 is missing
  15. Below figure 7: This whole explanation below figure 7 should be dropped since it is both conjectural and confusing. Differences can be attributed to different resistance. This can be due to different resistivity and to different geometry. But without further characterizations and statistical analysis it if not possible to say anything so detailed about the small differences between LIG and Cu/LIG

Author Response

Dear, Reviewer

  1. Introduction: Please comment that LIG is not really Graphene but just a conventional name for mainly 2D coordinated nanostructured carbon, as it is clearly demonstrated by Raman in figure 3

Carbon nanomaterials have been attracting significant attention in recent years due to their superior optical, electrical, physical, mechanical, and thermal properties and different types of carbon nanomaterials have been used in a variety of devices as active materials. Laser induced graphitization (LIG) has demonstrated its potentiality in spatially targeted modification of polymers into conductive carbon structure. Figure 3 shows the Both spectra have sharp symmertrical 2D peaks, providing that graphene in indeed produced during the LIG process

[1] Laser-induced graphene coated hollow-core fiber for humidity sensing doi.org/10.1016/j.snb.2022.131530

[2] Raman spectroscopy in graphene. doi.org/10.1016/j.physrep.2009.02.003

  1. Materials and methods: For easier reading please add the energy per pulse and peak power to table s1

Peak power

W

1466.67

Peak power density

W/

165992.72

We added the peak power and peak power density for easier reading as you mentioned.

  1. Materials and methods: “The porosity of this pattern was fabricated at the laser spot area”. The pattern was porous at....

We changed as you mentioned, thank you very much

  1. Results: "For fabrication of graphene based metallic hybrid materials, many methods...." Please explain briefly why the present method is better.

We added the details for why our present method is better

For fabrication of graphene based metallic hybrid materials, many methods are at-tempted such as reduction, liquid phase exfoliation and electrodeposition [27-29]. These methods can produce metal hybrid graphene with excellent performance, but they require harmful chemicals or an environment that applies high pressure and heat. We fabricate Cu oxide nanoparticles on the LIG surface very quickly and easily using LDW under ambient condition.

  1. Results: Figure 2 is difficult to understand. According to the text different rows (a-c, d-f, g-h) were made with different CuCl2 amounts, but they are presented at different magnifications. Moreover the substrate geometry on the first column looks different from the second and third. It is difficult to make sense of it. Probably the marker were misplaced, higher MAG from left to right and not from top to bottom

We changed figure configuration for easy understanding. And corrected marker misplaced as you mentioned. Thanks for pointing it out.

  1. Results, below figure 3: “The number of graphene layers can be identified by the ratio of the intensities of the 2D and G peaks” This statement is not correct for highly defective graphene, there are many papers about it Thickness of carbon nanostructures should be determined using TEM or SEM. (L. G. Cançado, A. Jorio, E. H. M. Ferreira, F. Stavale, C. A. Achete, R. B. Capaz, M. V. O. Moutinho, A. Lombardo, T. S. Kulmala, A. C. Ferrari, Nano Letters 2011, 11, 3190.A. Eckmann, A. Felten, I. Verzhbitskiy, R. Davey, C. Casiraghi, Physical Review B 2013, 88, 035426. )

As you mentioned, I removed the sentence above. thank you.

  1. Results: In the survey spectrum presented in figure 3 c, in the red curve it is possible to see also the Cl 2p peak at 200eV due to its incomplete removal. This peak should be also analysed.

Thanks for pointing out something we hadn't thought of. Cl is an anion in an aqueous solution of Cu. I lowered the importance because it was inevitably used to make Cu oxide nanoparticles. I apologize for not going into detail about Cl.

  1. Results below figure 4: “from 0.5–1.2 sun”, probably 0.5 to 12

We changed – to “to”. Thank u very much to point out.

  1. Whole section: The differences presented in figure 4 are very small and are apparently meaningless unless a statistical analysis of at least 3 samples per type is presented.

As you said, I checked the contents and confirmed that the difference was not large and meaningless, and conducted the experiment again with bare LIG fabricated according to the laser scanning speed and LIG with Cu oxide nanoparticles. Temperature difference showed a big difference depending on the presence and amount of Cu oxide. thanks for the good point.

  1. Results above figure 5: “This shows that Cu oxide not only produces heat more efficiently than bare LIG through photothermal con-version, but also provides heat resistivity to preserve heat [37].”: Please define what heat resistivity is, is it a decrease of thermal conductivity?

The text was not understood well, so I changed it as follows. thanks for the good point.

This shows that LIG with Cu oxide nanoparticles produces heat more efficiently than bare LIG through photothermal conversion.

  1. Results page 8: “These increased conjugated bonds result in a red shift of the absorption light spectrum, which greatly benefits the solar energy utilization efficiency” This statement should be demonstrated experimentally, by presenting absorption and reflection curves (using integrating spheres), else dropped

As you pointed out, the above sentence has been deleted because it is impossible to prove experimentally our lab. thank you.

  1. Results last 8 lines of 3.3 page 8 “the synergistic….” Overall the enhancement in performances seems too small to justify a different mechanism. Quite simply, the porous structure is obviously more absorbent (due to multiple reflections), and CuO particles are also black. The discussion is confusing and not supported by experimental evidence.

We changed “The synergistic effect of Cu oxide nanoparticles and LIG leads to enhanced optical absorp-tion and reinforced hot electron-phonon interactions at the interface [45]. This delicate synergistic effect can bestow photothermal hybridization materials with remarkably en-hanced photothermal properties” to Interface of Cu oxide nanoparticles and LIG leads to enhanced optical absorption and re-inforced hot electron-phonon interactions for efficient photothermal conversion

  1. Results 3.4 “However, the minimum operating volt-age of Cu/LIG heater in Figure 6c is about 0.5 V, which shows a 50% lower switching voltage than that of LIG heater.” This sentence is unclear, it should be either removed or better explained. If the IV curve is linear (figure 6a), there is no switching voltage and both devices can be operated at any voltage! The steps in figure 6b and 6c are arbitrary

thanks for the good point We removed the “In case of the LIG heater in Fig. 6b, the minimum operating voltage is 1 V. However, the minimum operating voltage of Cu/LIG heater in Figure 6c is about 0.5 V, which shows a 50% lower switching voltage than that of LIG heater. Additionally, Cu oxide alleviated the initial low increase behavior. Bare LIG has low increase section until 2.5 V, but Cu/LIG only go through it until 1 V. So,” as you said.

  1. Figure s4 is missing

We changed “s5” to “s4”

  1. Below figure 7: This whole explanation below figure 7 should be dropped since it is both conjectural and confusing. Differences can be attributed to different resistance. This can be due to different resistivity and to different geometry. But without further characterizations and statistical analysis it if not possible to say anything so detailed about the small differences between LIG and Cu/LIG

As you pointed out, we deleted the entire figure 7c part because we thought it was pointless to explain hypotheses that we couldn't prove experimentally. I deleted it in order to produce a high-quality paper. Thank you for helping me improve my thesis with 15 feedbacks.

Best Regards,

JUNUK LEE

Round 2

Reviewer 2 Report

Authors should clearly state in the manuscript or in the supporting information file the number of samples they have processed and the experimental statistics.

Author Response

Dear, reviewer

Authors should clearly state in the manuscript or in the supporting information file the number of samples they have processed and the experimental statistics.

We added the line "The excellent photothermal properties of Cu/LIG samples were observed as shown in Figure 4. We graphed the data as the average value of results of three or more experiments." 6page under caption of Figure 4.

Thanks for the nice point.
Thanks to you, the quality of my paper has improved.

In addition, the reference has been modified "2,3 and 31".

Best regards,

JUNUK LEE